# Symmetry of Hydrogen Bonds: Application of NMR Method of Isotopic Perturbation and Relevance of Solvatomers

**DOI:** 10.3390/molecules28114462

**Published:** 2023-05-31

**Authors:** Charles L. Perrin

**Affiliations:** Department of Chemistry & Biochemistry University of California, La Jolla, San Diego, CA 92093, USA; cperrin@ucsd.edu

**Keywords:** symmetry, low-barrier, shortness, hydrogen bonds, isotopic perturbation, NMR, disorder, solvation

## Abstract

Short, strong, symmetric, low-barrier hydrogen bonds (H-bonds) are thought to be of special significance. We have been searching for symmetric H-bonds by using the NMR technique of isotopic perturbation. Various dicarboxylate monoanions, aldehyde enols, diamines, enamines, acid–base complexes, and two sterically encumbered enols have been investigated. Among all of these, we have found only one example of a symmetric H-bond, in nitromalonamide enol, and all of the others are equilibrating mixtures of tautomers. The nearly universal lack of symmetry is attributed to the presence of these H-bonded species as a mixture of solvatomers, meaning isomers (or stereoisomers or tautomers) that differ in their solvation environment. The disorder of solvation renders the two donor atoms instantaneously inequivalent, whereupon the hydrogen attaches to the less well solvated donor. We therefore conclude that there is no special significance to short, strong, symmetric, low-barrier H-bonds. Moreover, they have no heightened stability or else they would have been more prevalent.

## 1. Introduction

Hydrogen bonding is a key feature of molecular structure and energetics [1,2]. It is key to understanding the structure and properties of water, proteins, and DNA, and it is currently of interest for understanding systems that exhibit molecular recognition [3].

In its simplest description, a hydrogen bond (H-bond) is the result of an electrostatic attraction between the positive end of an A–H dipole (A = O or N or F) and the negative charge in a lone pair on a neighboring electronegative atom B. Because an H atom is so small, it can approach close to that lone pair, making these interactions quite strong. The electrostatic forces then lead to a redistribution of the electron density, which may be interpreted as covalent bonding between H and B, or alternatively to the inclusion of an additional resonance form with a bond between the H and atom B (Equation (1)). 

A–H···:B ↔ A^–^:···H–B^+^(1)

What can be said about the structure of the H-bonded species? It may be expected that as the electrostatic interactions become stronger, H approaches closer to atom B while the A–H distance increases. Correspondingly, the barrier to H transfer from A to B decreases. In the limit of a vanishing barrier, the H-bond becomes symmetric, perhaps with equal A–H and H–B distances. The possible potential-energy diagrams are shown in Figure 1 (it should be recognized that a barrier that is at or below the zero-point energy for H motion, as in Figure 1c, leaves the H-bond as effectively symmetric). Moreover, a key requirement for symmetry is equality or near-equality of the acidity constants of the two donors AH and BH, for otherwise the H will bond more strongly to one than to the other, as in Figure 1a. Such asymmetry is much more common, so that the possibility of a symmetric H-bond, as in Figure 1b or Figure 1c, generally arises only when A = B and when the A–B distance is short, 2.4−2.5 Å for OHO H-bonds [4]. Of course, such an O–O distance then requires the O–H distances to be 1.2 Å, considerably greater than the usual 1.0 Å. It should be noted that this same question also arises for the so-called halogen bonds, where a halogen is bound between two donor atoms, but we leave that topic to experts in the field [5,6].

It must be remembered that the Uncertainty Principle limits our ability to specify the O–H or N–H distance. Because of zero-point energy, the expected value of the uncertainty in the square of that distance is *h*/4π√mk [7], where *h* is Planck’s constant, *m* is the mass of H, and *k* is the force constant for the vibration. For a 3200-cm^−1^ H-bond, that value corresponds to an rms uncertainty of 0.07 Å in the position of the H, which is small but not negligible compared to the difference between 1.2 and 1.0 Å. However, the question of the symmetry of the H-bond is not simply about the location of the H but about the potential-energy surface.

The possibility of such symmetry has been addressed in terms of the incursion of covalent nature into strong H-bonds [8]. As the proton affinities of the two donor atoms approach equality, covalency increases and the H-bonds become strong. The question of strength is then addressed by classifying the H-bonds as charge-assisted (further distinguished as neutral, anionic, or cationic) and/or resonance-assisted. 

It has become fashionable to recognize a group of H-bonds that are called “strong” or “low-barrier” and characterized by great strength, short distances, and a low or vanishing barrier to hydrogen transfer, as in Figure 1c. Admittedly the designation “short”, “strong”, or “low-barrier” depends on the criterion used for characterization. Al-though the energy of an ordinary H-bond is ca 20 kJ-mol^−1^, the strength of these low-barrier H-bonds (LBHBs) may be >50 kJ-mol^−1^ [9]. The designation “short, strong, low-barrier hydrogen bonds” has become a mantra to suggest a special importance to such H-bonds. In part, this was due to an influential article suggesting that LBHBs play a role in some enzyme-catalyzed reactions [10,11], although there were immediate challenges [12,13,14]. Such proposals continue to be offered [15,16,17], and rejected [18,19,20]. Although such H-bonds do have distinguishing characteristics, including long-range spin–spin coupling [21], and they are quite common in proteins (owing to compression), we have concluded that there is no need to invoke any extra stabilization due to the H-bond itself. In this review, we present our evidence.

This is in essence a question of the symmetry of the H-bond. Is the H centered between the two donor atoms? Or is it localized closer to one than to the other, although it may be jumping or tunneling rapidly from one to the other? The former possibility corresponds to the potential-energy diagram of Figure 1c, whereas the latter corresponds to Figure 1a,b. 

A centered H-bond is quite remarkable, since it requires >50 kJ-mol^−1^ to stretch a 3500 cm^−1^ O-H bond from the normal 1.0 Å to the 1.2 Å midway between the two oxygens. One rationalization for a compensating stabilization is that a symmetric H-bond has “covalent character” or else resonance assistance that confers extra strength [22].

How can such structures be distinguished? X-ray crystallography is ineffective, because it detects a time average, so a proton jumping back and forth in a double-well potential will appear symmetric. Infrared spectroscopy is complicated by the anharmonicity of vibrations, which makes interpretation difficult. Further, calculations are unreliable because the potential-energy surface is sensitive to small errors in the calculation and to subtle quantum effects of the low-mass H. The usual NMR methods fail because NMR is a slow form of spectroscopy, so even if there are two forms, they will interconvert on the NMR time scale. Fortunately, an NMR method is applicable, namely, the method of isotopic perturbation [23,24], which was developed by Martin Saunders and his coworkers and applied decisively to elucidating carbocation structures [25,26,27,28].

For many years, my laboratory has been addressing the question of the symmetry of H-bonds. This review is a summary of those studies, approximately in chronological order over a thirty-year period. The technique of isotopic perturbation is a subtle one, demanding careful attention to logic. The topic has also been reviewed by Shenderovich [29]. Several other articles tout the importance of short, strong, low-barrier H-bonds, but those often ignore the contrary evidence, including ours [30], and they do not deserve citation. 

## 2. Isotope Shifts

Our first approach to the symmetry of H-bonds focused on the detection of isotope shifts, as defined in Equation (2), in terms of the chemical shifts δ of a reporter nucleus X in the presence of either a heavier or a lighter isotope Y positioned *n* bonds away. For the special case of *n* = 0, this is a primary isotope shift, symbolized as *^P^*Δ, and the others, with *n* > 0, are called secondary. Poul Erik Hansen has conducted extensive studies of primary isotope shifts in H-bonded species [31,32], but our emphasis here is on secondary. It should be noted that conventions differ, but the definition we adopt focuses on the change from lighter to heavier Y, even though that usually results in values of Δ that are negative.
*^n^*Δ(X) = δ(X)_Yheavier_ − δ(X)_Ylighter_(2)

How can isotope shifts distinguish whether a species exists as a single symmetric structure or as two asymmetric tautomers in rapid equilibrium? There is always an intrinsic isotope shift *^n^*Δ_0_, arising simply because the reporter nucleus is affected by the isotopic substitution, but there is often a further shift *^n^*Δ_eq_ arising from the perturbation of an equilibrium. If the two tautomers are degenerate, of equal energy, then any chemical shift must be simply a 50:50 average. However, with isotopic substitution one tautomer can be favored. The isotopic perturbation introduces an equilibrium isotope shift, arising from differences in zero-point energy. Since ^13^C NMR is quite sensitive to the environment, small changes can be detected and taken as evidence for an equilibrium between two tautomers. One nice example is the enol–enethiol equilibrium in β-thiooxoketones [33].

We first focus on dicarboxylate monoanions, perturbed by ^18^O labeling in one of the carboxylates but not in the other. For a rapidly equilibrating mixture of such monoanions, the ^13^C NMR signal is averaged over two tautomers, which differ by whether the proton is on the ^16^O-labeled carboxyl or on the ^18^O one (strictly, there are four tautomers, and the proton can be on either O of a mono-^18^O-labeled carboxyl. This arbitrariness merely leads to a halving of the effect possible from double labeling at one carboxyl). The equilibrium constant between the two tautomers, *K*_T_, or the ratio of ^16^O and ^18^O acidity constants, is >1, owing to differences in zero-point energies [34,35,36,37]. The ^18^O-protonated tautomer is favored and thus weighted more heavily in the averaging. As a result, the ^13^C^16^O signal is shifted slightly downfield and the ^13^C^18^O slightly upfield. The isotope shift is related to *K*_T_ by Equation (3), where *D* is the difference between chemical shifts of carboxyl and carboxylate carbons in the monoanion Equation (4). Although those two shifts cannot be observed distinctly, *D* can be estimated from the chemical shifts of diacid and dianion Equation (5). Regardless of this approximation, the qualitative conclusion is that this method is capable of distinguishing symmetric H-bonds from asymmetric ones.
(3)$$\Delta_{\mathrm{eq}}=\Delta_{o}+D\;\frac{K_{T}-1}{K_{T}+1}$$
*D* = δ_COOH_ − δ_CO2_^−^(4)
*D* ≈ δ_diacid_ − δ_dianion_(5)

## 3. First Studies

Our first studies dealt with succinic (**1**), maleic (**2**), and phthalic (**3**) acid monoanions. These are classic cases of symmetric H-bonds, according to X-ray and neutron-diffraction studies of some of their crystals. Specifically, these anions exhibit a low barrier to H transfer, O−O distances of 2.4−2.5 Å [38], and a far-downfield ^1^H NMR signal near 20 ppm due to the internally H-bonded H [39]. X-ray and neutron-diffraction studies found that some crystals do exhibit centered H-bonds [40], although others do not, owing to different environments surrounding the two carboxyls [41]. 



To investigate whether those monoanions are symmetric in aqueous solution, they were prepared with exactly one ^18^O label [42]. This was easily achieved by hydrolyzing the acid anhydride in H_2_^18^O. The acid was then titrated in the NMR probe with aliquots of KOH. The carboxyl carbons of the diacid or dianion show a small ^18^O-induced intrinsic isotope shift Δ_o_ of −26 ppb (note that these small isotope shifts are measured in ppb, not the customary ppm). The isotope shift, ^1^Δ(^13^C) of ^13^C at natural abundance, rises to a maximum in the monoanion and reverts to the intrinsic isotope shift in the dianion. The results are shown in Figure 2 [42]. 

Those data show that the isotope shifts of the monoanions are larger, at both carboxyl and ipso carbons (where ipso designates the aromatic carbon that bears a substituent [43]) than in either the diacid or dianion. This was firm evidence, supported by four control experiments (effect of D_2_O, temperature dependence, comparison of mono-^18^O- and di-^18^O-labeled carboxyls, isotope shifts at distant carbons), for a tautomeric equilibrium and for asymmetric H-bonds. The results on hydrogen maleate and phthalate were especially surprising since these monoanions can show symmetric H-bonds in crystals. 

The data could be analyzed to obtain isotope effects *K*_a_(^16^O)/*K*_a_(^18^O) of 1.0060 ± 0.0001, 1.0044 ± 0.0001, and 1.00926 ± 0.0009 for aqueous succinic, maleic, and phthalic acids, respectively. These values agree well with the (double) isotope effect of 1.0117 on the acidity of formic acid-^18^O_2_ [44]. The key result is that maleic and phthalic acids show the same isotope effect as succinic. Because succinate monoanion has no internal H-bond it must exist as two tautomers in an equilibrium that can be perturbed by ^18^O substitution. The observation that maleate and phthalate monoanions show the same perturbation means that they too exist as two tautomers, with the H in the internal H-bond bonded more strongly to one or the other of the oxygens and not symmetrically placed between them. These anions are asymmetric in aqueous solution, in contrast to their symmetry in the crystal. This is a simple counterexample to the hope that a crystal structure describes the structure in solution. 

In a follow-up study, the observed isotope shift Δ of the monoanion of mono-^18^O-phthalic acid (**3**) was observed to vary with temperature, being higher at lower temperature [45]. This behavior is seen not only at the carboxyl carbon but also at the ipso carbon. This is strong confirmatory evidence for isotopic perturbation of an equilibrium. However, in DMSO-*d*_6_, CD_3_CN, and THF-*d*_8_, neither maleate (**2**) nor phthalate (**3**) monoanion showed an isotope shift perceptibly larger than the intrinsic shift. Therefore, in these nonaqueous solvents, the H seems to be centered in the H-bond, and there is no equilibrium between asymmetric tautomers. In THF-*d*_8_, even succinate (**1**) monoanion showed no increased isotope shift, meaning that it has taken a cyclic conformation, to permit an H-bond that is not only intramolecular but also symmetric. In such a nonpolar solvent, the best solvation for the carboxylate anion is H-bonding to the carboxylic-acid group at the other end of the ion.

This apparent contrast between water and nonaqueous solvents was attributed to the special nature of water. Water H-bonds to the carboxyl groups in such manner that on average both carboxyls are equally solvated. However, water is a disorganized solvent, and it would require considerable negative entropy for both carboxyls to be identically solvated. If instead one carboxyl group happens to be more strongly solvated than the other, it becomes the carboxylate and the H attaches to the other carboxyl. Then, as the solvent fluctuates, the proton oscillates between the two carboxyls, but at any instant the H-bond itself is asymmetric. Indeed, there are well-known examples of maleate (**2**) monoanions with asymmetrically placed counterions that show an asymmetric H-bond [46,47]. In contrast, in aprotic organic solvents, the H-bonds of maleic, phthalic, and succinic acid monoanions seemed to be symmetric, because those solvents solvate an anionic carboxylate group less well than they solvate the delocalized negative charge of a symmetric anion. It was concluded that these monoanions have a symmetric H-bond in organic solvents. This conclusion was generally accepted, since it agreed with some previous results [48]. 



However, on further scrutiny, the perturbation shift Δ_o_−Δ in organic solvents is recognized as quite small but never really zero. It might again be evidence for the perturbation of an equilibrium. Therefore, the symmetry of the H-bonds in monoanions of a wider variety of dicarboxylic acids was explored, including cyclopentene-1,2-dicarboxylate (**4**), furan-3,4-dicarboxylate (**5**), bicyclo[2.2.1]-heptadiene- 2,3-dicarboxylate (**6**), cis-cyclopropane-1,2-dicarboxylate (**7**), dimethylmaleate (**8**), and cyclohexene-1,2-dicarboxylate (**9**) monoanions. 

Table 1 lists measured isotope shifts for these diacid monoanions in DMSO-*d*_6_ [49]. Although the carboxyl shift of a monoanion might earlier have seemed to be the same as the intrinsic shift Δ_o_, the average difference Δ_o_−Δ_HA−_ is 1.5 ppb and no value is <0. The average ipso shift is 13 ppb, significantly greater than the intrinsic shift of ≤4 ppb seen in the diacid. In summary, all isotope shifts in the monoanions are larger in magnitude than the corresponding intrinsic shift. Similar results were seen in THF-*d*_8_. Moreover, for phthalate (**3**) monoanion in CD_2_Cl_2_ at −55 °C, the measured Δ_o_−Δ_HA−_ is larger, 23 ppb, as expected for the temperature effect on an equilibrium.

For phthalate (**3**), the isotope shift at the carboxyl carbon is hardly greater than the intrinsic shift. Nevertheless, the isotope shift at the ipso carbon is 13–15 ppb, significantly larger than its intrinsic shift. This result holds for all the organic solvents examined. Therefore, we conclude that phthalate monoanion does not exist as a single symmetric ion, a conclusion opposite to our original one for organic solvents. Moreover, the comparison suggests that carboxyl isotope shifts are inadequate to determine the symmetry of its H-bond. The ipso signals seem to be more sensitive to the state of protonation of the carboxyl group and thus more diagnostic. More importantly, there is no resolvable intrinsic shift at the ipso carbon. Therefore, the large ipso shift in the monoanion is indisputable evidence for a tautomeric equilibrium, perturbed by ^18^O. All of these monoanions have asymmetric H-bonds in all solvents [49]. Similar conclusions apply to the monoanions of succinic, methylsuccinic, and both stereoisomers of 2,3-dimethylsuccinic acids [50]. 

If the disorder of aqueous solvation is responsible for the asymmetry of these H-bonds in water, then why are they also asymmetric in organic solvents? It must be recognized that these monoanions are paired with counterions in nonpolar solvents. If that counterion is placed asymmetrically relative to the two carboxyls, then one tautomer will be favored. This asymmetry has the same consequence as unequal H-bonding to the two carboxyls in water.

## 4. Isotopic Perturbation by Deuterium

The intramolecular H-bonds in the enol forms of malonaldehyde (**10**) and acetylacetone (**11**) are strong. They are considered to be resonance-assisted H-bonds, stabilized by shortening the O–O distance and lengthening the O-H bond [51,52]. However, it must be recognized that the H-bond resides in the σ system, whereas resonance is a π phenomenon. Besides, the symmetry of such H-bonds is less certain. Ab initio calculations must calculate the O–O distance accurately and must take adequate account of both electron correlation and zero-point energy. One review concluded that double-well potential is supported by experimental data but that calculations do not give a reliable energy barrier [53].



This is a favorable system for the application of the method of isotopic perturbation to experimentally distinguish a single-well potential from a double-well one. In contrast to the previous examples, these enols are not anions, so solvation effects are much less significant. More important is the possibility of using deuterium as the perturbing isotope, since isotope effects of deuterium are much larger than for ^18^O. 

We chose to study the enol of 2-phenylmalonaldehyde (**12**), with a phenyl at C2 for ease of synthesis and to avoid the configurational flexibility around the C=C of **10** [54]. If the OH hydrogen of **12** is in a double-well potential, then there are two asymmetric monodeutero tautomers **12**-*d*_1_(enol) and **12**-*d*_1_(ald) in equilibrium, differing in whether the deuterium is on the enol or aldehyde carbon. The monodeuterium substitution perturbs the equilibrium between them, whereas there is no perturbation for isotopologs **12**-*h*_2_ and **12**-*d*_2_, which serve as a comparison. 



If **12** is a mixture of two asymmetric tautomers, then it can be shown that the equilibrium isotope shift is given by Equation (6), where *K* is the tautomeric equilibrium constant [**12**-*d*_1_(enol)]/[**12**-*d*_1_(ald)] and *D* is the difference between ^13^C NMR chemical shifts of =CH-O and –CH=O. This difference can readily be estimated by comparing the chemical shift of **12**-*h*_2_, where the equilibrium constant is 1, with the weighted average of the chemical shifts for =CH-O and -CH=O, respectively, in **12**-*d*_1_(enol) and **12**-*d*_1_(ald).
(6)$$\Delta_{\mathrm{eq}}=\delta_{\mathrm{CHD}}-\delta_{\mathrm{CHH}}=\frac{1}{2}\frac{K-1}{K+1}\;D$$

The resulting isotope shift can be estimated, for both ^13^C NMR and ^1^H NMR. The C-H stretching frequency of an enol is 3020 cm^−1^, whereas that of an aldehyde is much lower, 2770 cm^−1^ [55]. The zero-point energy for **12**-*d*_1_(enol) is then 1/2[2770 + (1/√2)3020] cm^−1^ and that for **12**-*d*_1_(ald) is 1/2[3020 + (1/√2)2770] cm^−1^. These values then lead to an energy difference of 37 cm^−1^, corresponding to an equilibrium constant *K* of 1.2 at 25 °C. The equilibrium favors the tautomer **12**-*d*_1_(ald), with H on the lower-frequency CH=O bond and D on the higher. Moreover, the separation between aldehyde and enol ^13^C chemical shifts is ca. 20 ppm, as judged from the chemical shifts of **12** in the solid state [56]. Therefore, Δ_eq_ is expected to be ca. +1 ppm in the ^13^C NMR. Similarly, in the 500-MHz ^1^H NMR spectrum, where the separation between chemical shifts of =CH-O and -CH=O is ca. 2 ppm, we may expect a Δ_eq_ of ca. 0.1 ppm. Note that the deuterium substitution must be on carbon, not in the OH H-bond, which would not be informative for this distinction. 

Figure 3 shows the carbonyl region in the ^1^H- and ^2^H-decoupled ^13^C NMR spectrum of a mixture of **12**, **12**-*d*_1_, and **12**-*d*_2_ in CDCl_3_ at −37.9 °C [54]. The signal furthest downfield (at the left) can be assigned as the CH of **12**-*d*_1_ and the most intense signal as the CH of **12**-*d*_0_. These remain the same even without ^2^H decoupling. The other two signals are from carbons attached to a deuterium. On the basis of the relative amounts, the taller of these can be assigned as the CD carbon signal of **12**-*d*_1_. The weakest signal, not always visible, is assigned as **12**-*d*_2_.

The chemical-shift difference between **12** and **12**-*d*_2_ in Figure 3 is −261 ppb. This is the sum of two intrinsic isotope shifts, ^1^Δ_0_ + ^3^Δ_0_. Because this is so large, we must compare **12**-*d*_1_ with undeuterated **12**-*d*_0_, rather than comparing the two carbons of **12**-*d*_1_. The chemical-shift difference between **12**-*d*_1_ and **12**-*d*_0_, or equivalently between the deuterium-bearing carbons of **12**-*d*_2_ and **12**-*d*_1_, is large, 759 ppb in either CDCl_3_ or C_6_D_6_. This value represents the sum ^1^Δ_eq_ + ^3^Δ_o_. Similarly, the peak separation in the ^1^H NMR spectrum is 50 ppb, which is Δ_eq_ + ^4^Δ_o_. Such large separations are evidence for asymmetric H-bonds, even in these nonaqueous solvents. 

The temperature dependence of the isotope shifts for **12** in CDCl_3_ is revealing [54]. The sum ^1^Δ_o_ + ^3^Δ_o_ of intrinsic isotope shifts is temperature-independent. In contrast, the magnitudes of both ^1^Δ_eq_ + ^3^Δ_o_ from the ^13^C spectrum and Δ_eq_ + ^4^Δ_o_ from the ^1^H spectrum increase with decreasing temperature. The slope of ^1^Δ_eq_ + ^3^Δ vs 1/*T* is 202 ppb-K. If the chemical-shift difference between -CH=O and =CH-O chemical shifts is taken as 20 ppm, Δ*H*° is then 27 cm^−1^, in decent agreement with the 37-cm^−1^ difference in zero-point energy estimated above for the two tautomers. Likewise, the ^1^H isotope shift of +50 ppb at 20.7 °C or +60 ppb at −37.9 °C is in decent agreement with the 0.1 ppm estimated above. 

It must be noted that the isotopic perturbation itself is not what creates the asymmetry that we have inferred. There are actually two such perturbations, not only the D substitution but also the rare ^13^C needed for ^13^C NMR. Nevertheless, according to the Born–Oppenheimer approximation [57], the potential-energy surface governing nuclear motion is determined only by the electrons and must be independent of nuclear masses. Therefore, isotopic substitution is fully capable of distinguishing single-well from double-well potentials. That was not really a question for **12**, but this study demonstrates the power of the method. 

A related question is whether a metal chelate is symmetric, with the motion of the metal described by a single-well potential, or asymmetric, in a double-well potential. According to crystal structures, bidentate metal β-diketonates are firmly believed to be symmetric [58]. However, this is not conclusive because apparently equal M-O bond distances may be only the time average of a mixture of asymmetric structures. The molecular symmetry of ML*_n_* (M = Li, Na, K, Al, Pd, Rh, Si, Sn, Ge, Sb, etc., L = anion of 2-phenylmalondialdehyde) in solution was therefore probed by that same method of isotopic perturbation [59]. A statistical mixture of 2-phenylmalondialdehyde-*d*_0_ (**12**), -1-*d*, and -1,3-*d*_2_ was synthesized and converted to various metal complexes. For LiL, NaL, and KL, the ^13^C NMR isotope shifts ∆ (= δ_CH(D)_−δ_CH(H)_) for the aldehydic CH are small and negative, consistent with a free, uncoordinated L^–^ that is a resonance hybrid. Some complexes of silicon showed two separate aldehydic signals, which means that their ligands are monodentate. The isotope shifts are small and positive for AlL_3_, PdL_2_, Rh(CO)_2_L, SiX_3_L, SiL_3_^+^X^–^, (CF_3_)_3_GeL, SbCl_4_L, (EtO)_4_TaL, and (EtO)_4_NbL. Positive isotope shifts are unusual, but since they are small and temperature-independent, they must be intrinsic and indicative of metal chelates that are symmetric, with the metal fixed in the center between the two oxygens. In contrast, large positive isotope shifts were observed for Ph_3_GeL, Me_3_GeL, Bu_3_SnL, and Ph_4_SbL, which consequently must be asymmetric. However, it is likely that these are monodentate complexes undergoing rapid metal migration. NMR experiments indicated an intermolecular mechanism for exchange, involving a double metal transfer. The main conclusion though is that we have found no example of a metal chelate that is asymmetric, confirming the long-held belief and in contrast to the asymmetry of the H-bond in **12**. Of course, a metal chelate ought to be symmetric, because the metal has sufficient orbitals to form bonds to both donor oxygens, in contrast to hydrogen, which has only one.

## 5. NMR Titration. A Brief Digression

The examples above demonstrate the ability of NMR instrumentation to measure small isotope shifts. That sensitivity then stimulated the development of an NMR titration method for measurement of the difference in acid dissociation constants of two or more structurally similar compounds [60]. The power of the method was demonstrated by measuring the relative acidities of the two stereoisomers of 4-t-butylcyclohexylamine (**13**), the relative basicities of 4-*t*-butylcyclohexanecarboxylic acid (**14**), and the relative basicities of all four stereoisomers of 2-decalylamine (**15**) in a single ^1^H NMR titration. The method is capable of exceptionally high precision and accuracy, and it can be carried out without the necessity of measuring pH. 



Among further substrates subjected to NMR titration were the stereoisomeric *N*-phenylcyclohexylimidazoles (**16**) [61], glucosylimidazoles (**17**, R = H, CH_3_CO) [62], *N*-(glucopyranosyl)anilines (**18**, R = CH_3_) and *N*-(4-*t*-butylcyclohexyl)anilines (**19**) [63], and the series of cycloalkylamines (**20**, *n* = 3–12, 15–16, 21) [64]. It could further be shown that the deuterium isotope effects on the basicity of triethylamine are nonadditive, demonstrating an isotope effect on an isotope effect [65], which is a counterexample to the Rule of the Geometric Mean [66]. It may be noted that linear least-squares analysis of one of those NMR titrations gave a correlation coefficient of 0.999999. A detailed guide to the method has been published [67].



## 6. Symmetry of NHN H-Bonds in Proton Sponges

The intramolecular NHN H-bonds of protonated 1,8-bis(dimethylamino)naphthalene (**21**) and 2,7-dimethoxy-1,8-bis(dimethylamino)naphthalene (**22**) are of interest. The corresponding diamines are called Proton Sponges because they are strongly basic, with basicities enhanced by 7.5 and 11.5 pK units, respectively, relative to *N*,*N*-dimethylaniline [68,69]. Such enhancements have been attributed to a strengthening of the H-bond by 40-60 kJ-mol^−1^. 



Why are their H-bonds so strong? Is it because they are LBHBs, where the barrier to H transfer is so low that the H-bonds have become symmetric? If so, **21** and **22** would exist as a single structure (Figure 4a), rather than as a pair of interconverting tautomers (Figure 4b). These are certainly good candidates for symmetric H-bonds, according to a variety of criteria: the NH chemical shift is far downfield, at δ 18.46 [70], a feature that has been considered most unambiguous for characterizing LBHBs [71]. The nitrogens are forced into proximity, and the short N–N distance favors a single-well potential, as in Figure 4a. Besides, the *N*-methyls preclude NH H-bonds to solvent.

Again, the method of isotopic perturbation can distinguish between the two possibilities in Figure 4. Again, deuterium can produce the largest ^13^C NMR isotope shifts, but in this case by replacing the *N*-CH_3_ by *N*-CD_3_. Again, there is an intrinsic contribution *^n^*Δ_o_, which is upfield and falls off rapidly with *n*, the number of bonds between the reporter carbon and the deuteriums. Moreover, if the H-bond in **21** or **22** is asymmetric, as in Figure 4b, then there is an additional contribution of the *N*-CD_3_ substitution to Δ_obs_. α-deuteration perturbs the tautomeric equilibrium because it increases amine basicity [72,73] by a factor of 1.105, as modeled by the comparison of PhN(CH_3_)_2_ and PhN(CH_3_)(CD_3_). This results in a perturbation isotope shift Δ_eq_ again given by Equation (6), but where *D* = δ_BH+_ − δ_B_, the chemical-shift difference between exchange-related carbons proximal and distal to the NH^+^ in a static tautomer, as modeled by a comparison of chemical shifts in PhNH(CH_3_)_2_^+^ and PhN(CH_3_)_2_. A further advantage is that the mixture provides numerous signals from the four pairs of ring carbons, whereas the two bridgehead carbons (C9 and C10) common to both rings can experience only an intrinsic shift, proportional to the total number of CD_3_ groups. 

To apply the method of isotopic perturbation, a statistical mixture of all the isotopologs of both **21** and **22** was prepared by methylating 1,8-diaminonaphthalene or 2,7-dimethoxy-1,8-diaminonaphthalene with a 50:50 mixture of dimethyl sulfate and dimethyl sulfate-*d*_6_ and converting the free base to its thiocyanate or tetrafluoroborate salt [74]. Figure 5 shows the C1,8 region of the ^13^C NMR spectrum of **21**·HSCN in DMSO-d_6_, along with the assignments of intrinsic and perturbation isotope shifts. The *d*_0_ signal was assigned by adding authentic **21**-*d*_0_. 

Table 2 lists the isotope shifts of the various ^13^C NMR signals of **21**·HSCN and **22**·HBF_4_ in DMSO-*d*_6_. For **21**, both the *N*-methyl and C9 show only intrinsic shifts. There is even a detectable Δ_0_ at C4,5. In contrast, the quintets for C2,7 and C3,6 must be attributed to perturbation shifts, because *d*_0_ is the central peak of each. The signals of C1,8 and C4,5 exhibit both intrinsic and perturbation isotope shifts, since their signals due to *d*_0_ are neither central nor an extremity. The behavior of **22** is quite similar. The quintet for C2,7 is due to a perturbation shift Δ_eq_. Signals of C4,5 exhibit both perturbation and intrinsic shifts. A minor difference is that isotope shifts at C1,8 and C3,6 are unresolvable. 

The key result is that the observations of perturbation isotope shifts Δ_eq_ at all four ring carbons of **21**·H^+^ and at two carbons of **22**·H^+^ indicate that the isotopes perturb a tautomeric equilibrium. It was concluded that in solution, the H-bonded proton must reside in a double-well potential surface (although see below), and the ion must exist as a pair of rapidly converting tautomers. This conclusion is consistent with evidence from neutron scattering, vibrational spectroscopy, and computation [75]. These ions are unequivocally asymmetric. Even though the N–N distance is short enough to allow a single-well potential, the NHN path is a longer arc. The role of an NHN angle < 180° has been noted as favoring an asymmetric H-bond [76]. 

The *N*-methyls of **22**·H^+^ (but not of **21**·H^+^) show an unusual stereospecificity across their H-bond [74]. Whereas the *N*-methyl signals of **21**·H^+^ show a 78-ppb doublet, **22**·H^+^ shows a doublet of doublets, corresponding to two unequal intrinsic shifts of -80 and −25 ppb. The most downfield signal is due to the *d*_0_ isotopolog. The 80 ppb is consistent with a ^3^Δ_0_ from a geminal CD_3_, as also seen in **21**·H^+^. The other splitting is remarkable in that it is a doublet, not a triplet. Therefore, only one of the methyls on the other N can be responsible for this ^5^Δ_0_, while deuteration at the other position has no effect. This requires that cis/trans relationships be preserved on the NMR timescale. If not, both distal methyls would have produced isotope shifts, and a pair of triplets would have resulted. However, this would require C_aryl_–N rotation, which could be slow in this congested environment because it also requires breaking of the H-bond. Indeed, rotation of a similar dialkylamino group is slow in N,N′-dibenzyl-N,N′-dimethyl-1,8- naphthalenediamine·H^+^ [77]. 

Equal basicity of the two nitrogens is a necessary condition for a symmetric H-bond. Yet, the essence of the isotopic perturbation method is the use of CD_3_ groups to render the basicities unequal. Might that isotopic substitution, as well as the substitution of ^13^C for ^12^C, have destroyed a symmetry that would have been present without isotopes? The answer, as discussed above for **12**, is that the isotopic substitution itself cannot have converted a single-well potential into a double-well one.

Why then are bis(dimethylamino)naphthalenes (“Proton Sponges”) so basic? Relative to *N*,*N*-dimethylaniline, the basicity of **21** is enhanced 10^8^-fold and 10^12^-fold for **22**. If **21**·H^+^ and **22**·H^+^ are not symmetric, then they are not stabilized by resonance involving two identical resonance forms. We therefore conclude that the H-bonds are not strong in themselves. Instead, we return to the long-standing interpretation that the basicity of these diamines arises from relief of strain upon protonation [78,79]. 

Figure 6a illustrates how relief of strain can make the H-bond appear to be unusually strong [74]. For a model base, B_0_, H-bond formation with HA is assumed to lower the energy by Δ*G*°_HB°_. If another base, B, is destabilized by an additional ΔG°_destab_, then H-bond formation with that same HA, with relief of that destabilization, will lower the energy by ΔG°_HB_. This stabilization is greater than ΔG°_HB°_, but not because of a stronger H-bond.

## 7. Is There Any Role for LBHBs in Enzyme Catalysis?

It is certainly true that low-barrier H-bonds have attracted great interest for their possible role in stabilizing intermediates or transition states in enzyme-catalyzed reactions, including recent evidence from neutron crystallography that the deuterium in an aspartate aminotransferase is equidistant between two donor atoms [80]. A role for low-barrier H-bonds in green fluorescent protein has been rejected, despite a very short O–O distance [81], and this critique has been authoritatively reviewed [82].

In accounting for the role of unusual H-bonds in some enzymatic catalysis, there may be no need to invoke an LBHB or any extra stabilization due to the H-bond itself. Figure 6b illustrates how relief of strain can lower activation energy [74]. It parallels Figure 6a, but it also resembles a figure attributing an increased H-bond strength to the greater variability of p*K*_a_ in DMSO [83]. If a basic group B^–^ forms an H-bond in water with acid HA, the energy would be lowered by Δ*G*°_aqHB_. However, if B^–^ in an enzyme active site is destabilized by Δ*G*°_destab_, then formation of an H-bond with HA lowers the energy by Δ*G*°_enzHB_, an amount greater than Δ*G*°_aqHB_ (the destabilization of B^–^ can arise either by forcing its lone pairs to overlap with other lone pairs or by placing it in an aprotic environment, where it cannot form H-bonds. The first possibility is like **21** and **22** and the second is justified by the observation that H-bonds appear stronger in aprotic solvents). However, the stabilization is not due to any unusual strength of the H-bond itself but to relief of a destabilization. The extra stabilization of Δ*G*°_enzHB_, over that of Δ*G*°_aqHB_, can then lower the transition-state energy. Although enzymes usually decrease activation energy by stabilizing the transition state, it is not a new idea that they may alternatively do so by introducing strain into the enzyme–substrate complex [84]. In the context of LBHBs, this requires that the energy of substrate binding be sufficient to desolvate an anionic group or to force it against another anion. If, on passing to the transition state, the strain is relieved by inserting a proton and forming an H-bond, then *k*_cat_ can be increased. The stabilization is then not due to any unusual strength of the H-bond itself but to relief of strain.

## 8. Further NHN and OHO H-Bonds

Hydrogen bonds analogous to the NHN H-bonds of cationic **21** and **22** are found in neutral enaminoimines **23** (Ar = Ph or 3,5-Xyl), as well as in **24**, with an OHO H-bond. These have seven-membered rings, which may compress the N–N or O–O distance and favor short H-bonds. That N–N interaction is unusually strong, as indicated by an ^15^N-^15^N coupling across the N-H-N in an asymmetric analog [85,86]. To test for the possible symmetry of their H-bonds, they can be perturbed by deuterium substitution, as in **25** (X = O or NAr). 



For **25** (X = NPh) in DMSO-*d*_6_, the aldehydic CH is shifted downfield by 223 ppb, while the CD is shifted upfield by 542 ppb [87]. For **25** (X = N-3,5-Xyl) in CDCl_3_, these values are 176 and 249, respectively. Further, for **24** in CDCl_3_, they are 200 and 279 ppb. These are enormous isotope shifts, dominated by large intrinsic shifts, but also showing large perturbation shifts owing to the substantial differences in both zero-point energies and chemical shifts between imine or aldehyde and enamine or enol. Thus, each of these species exists as a pair of rapidly interconverting tautomers.

It follows that the symmetry of the H-bond is not determined simply by the N–N or O–O distance. Even though that distance is short in these seven-membered rings, that shortness does not lead to a symmetric H-bond.

## 9. Role of Solvent Disorder

A crucial question persists. Why are symmetric H-bonds observed in crystalline phases but not in solution? The reason is not because of polarity, inasmuch as a crystal is also highly polar, with strong electric fields due to nearby ions. We proposed that the difference is because the local environment in solution is disordered. If the O or N atoms in the H-bond are subject to unequal solvation, this can induce an asymmetry in the H-bond. The resulting disorder can then be responsible for the observed asymmetry in solution, whereas a crystal can guarantee a symmetric environment. Indeed, this distinction is supported by computations [88,89,90].

Does the observation of isotope shifts even require a double-well potential? There are certainly two tautomers, since the equilibrium between them can be perturbed by isotopic substitution. Each individual tautomer must be asymmetric. However, this does not require that the potential-energy surface be double-well. Even if the potential is intrinsically single-well, as in Figure 7a, preferential solvation of one end of the H-bond could stabilize one tautomer more than the other [91]. That would change the potential either to that of Figure 7b or to that of Figure 7c, or to more or less extreme versions of these potentials. At any instant, the H-bond would be asymmetric. However, as the solvent reorganizes, the potential switches between one like Figure 7b and one like Figure 7c. Such a single-well potential would permit an equilibrium between two tautomers, detectable through the ability of isotopic substitution to perturb that equilibrium. 

## 10. Acid–Base Complexes

Might a complex between a neutral acid AH and a neutral base B show a centered H-bond? Of course, since A and B are different, such an H-bond, even if centered, cannot be symmetric. An advantage though is that the interaction is intermolecular, so that there are no geometric constraints imposed on the A–B distance, allowing the H-bond to adjust to its most stable configuration.

Figure 8 presents this approach. The potential-energy surface for a pyridine-dichloroacetic acid complex (**26a**, **26b**, R = CHCl_2_) changes with the basicity of the pyridine. As the basicity increases, the N–H distance decreases and the O–H distance increases, so that the structure shifts from **26a** toward **26b**. At some intermediate stage, the pyridine basicity matches that of the carboxylate, and the double-well potential becomes effectively symmetric, even though N and O are different. At this point, a transition to a single-well potential becomes possible. Indeed, a centered hydrogen was seen by neutron diffraction in 4-picoline–pentachlorophenol at 90 K [92]. Low temperature may be needed, if a centered hydrogen is associated with a negative entropy. The effect of temperature on H-bond structure has been well documented [93].

Again, isotopic perturbation can distinguish a tautomeric mixture (**26a** + **26b**) from a single species with a shared hydrogen. The perturbation is achieved by two ^18^Os in the carboxyl group, which reduce the acidity of RCOOH and shift the equilibrium toward **26b**. The isotope shift is detected as the chemical shift of the carboxyl carbon of the ^18^O_2_ acid, relative to that of the ^16^O_2_ acid [94]. The observed isotope shift is the sum of an intrinsic shift ^1^Δ_0_ and a shift ^1^Δ_eq_ due to perturbation of the equilibrium between the two tautomers. This latter can be shown to be given by Equation (7), where δ_OH_ and δ_O−_ are the chemical shifts of the carboxyl carbons in tautomers **26a** and **26b**, respectively, *K*_e_ is the equilibrium constant [**26b**]/[**26a**], averaged over ^16^O and ^18^O acids, and *K* is the ratio *K*_a_(^16^O)/*K*_a_(^18^O), which is ~1.02 [95]. This ^1^Δ_eq_ approaches zero for large or small *K*_e_ and reaches a maximum when *K*_e_ = 1. However, if the double-well potential of separate tautomers **26a** and **26b** becomes single-well, then there will be only a single species, and ^1^Δ_eq_ will disappear.
(7)$${\;}^{1}\Delta_{\mathrm{eq}}=(\delta_{\mathrm{OH}}-\delta_{O-})\;\frac{K_{-}-1}{1+K+K^{1/2}Ke+K^{1/2}/Ke}$$

Experimentally, at −43 °C and −81 °C in CD_2_Cl_2_, the ^18^O-induced ^13^C NMR isotope shifts of the carboxyl carbons in 1:1 complexes of dichloroacetic acid with a series of pyridines show maxima [94]. The increase in isotope shift is due to perturbation of a closely balanced tautomeric equilibrium between **26a** and **26b**. Therefore, each of these complexes is a mixture of two tautomers, not a single structure with a single-well-potential hydrogen bond. However, the complexes of pyridine and 3-picoline were exceptions whose isotope shifts seemed to revert to the intrinsic ^1^Δ_0_. We therefore tentatively concluded that each of those two complexes is a single structure, with its hydrogen shared between nitrogen and oxygen. 

To investigate this issue more thoroughly, ^18^O-induced ^13^C NMR isotope shifts in 1:1 complexes of dichloroacetic acid with a series of nine different pyridines, including pyridine itself and 3-picoline, in CD_2_Cl_2_, were measured at −43 °C and −81 °C [96]. At these low temperatures, the OH chemical shifts approach 20 ppm, consistent with strong H-bonds. The data are shown in Figure 9. They fit Equation (7) quite well, with no reversion of any observed isotope shift to the intrinsic Δ_o_ at intermediate p*K*_a_, when the basicities of the pyridine and the carboxylate match. Therefore, each of these complexes is a mixture of two tautomers. This conclusion is contrary to the previous tentative one, and it shows that the H-bond in these complexes never becomes centered in a single well-potential.

## 11. Solvatomers

NMR studies as described above showed that the monoanions of many mono-^18^O-labeled dicarboxylic acids, including maleate **2** and phthalate **3**, exist as a pair of tautomers in aqueous solution. Although initial results had suggested that they become symmetric in organic solvents, further studies showed that they remain a pair of tautomers in all solvents, each with an asymmetric H-bond. This asymmetry in solution contrasts with the symmetry that is calculated in the gas phase and observed in some crystals. 

It was therefore proposed that the asymmetry is due to the disorder of the local environment, which prevents identical solvent or counterion interactions with both of the carboxyl groups. In particular, identical H-bonding by water or other protic solvent to both carboxyl groups is unlikely, because this would require a network of H-bonds organized through the solvent. Likewise, in less polar solvents, an associated counterion cannot be localized symmetrically with respect to both carboxyls. 

The role of the counterion in disrupting the symmetry of the H-bond in phthalate monoanion may be addressed with zwitterion **27** (R = *n*C_4_H_9_ or *n*C_8_H_17_) [97]. The two oxygens must have identical basicity, and they are forced close to each other, so that a symmetric H-bond is possible. The cationic nitrogen is fixed equidistant from the two carboxyls. This symmetric location eliminates the dynamic disorder of a separate countercation relative to the negative charge. Introduction of one ^18^O then allows the nature of its H-bond to be probed by the method of isotopic perturbation.



In CD_2_Cl_2_, the ^1^H NMR spectrum of **27** (R = *n*C_8_H_17_) shows a signal far downfield, at δ 20.08, characteristic of a low-barrier H-bond. ^18^O-induced ^13^C isotope shifts of zwitterion **27** (R = *n*C_4_H_9_ or *n*C_8_H_17_) and of its cationic diacid in CD_3_OD are collected in Table 3. Just as with phthalic acid, the carboxylic-acid carbons show an intrinsic isotope shift of 25–26 ppb for the protonated cation of **27** (R = *n*C_8_H_17_) in both CD_3_OD and CD_2_Cl_2_ and for the protonated cation of **27** (R = *n*C_4_H_9_) in CD_3_OD, but the intrinsic isotope shift at the ipso carbons could barely be resolved. 

The magnitude of the equilibrium isotope shift Δ_eq_ was obtained by subtracting the intrinsic isotope shift Δ_0_ observed in the cationic diacid from the observed shift Δ_obs_ in the zwitterion **27**. A Δ_eq_ of 15 ppb is seen for the carboxyl carbons of both **27** (R = *n*C_4_H_9_) and **27** (R = *n*C_8_H_17_) in CD_3_OD. An even larger Δ_eq_ is seen at the ipso carbons. In CD_2_Cl_2_, although the Δ_eq_ of 2 ppb at the carboxyl carbon of **27** (R = *n*C_8_H_17_) is small, hardly above the resolution, nevertheless, Δ_eq_ at the ipso carbon is 7 ppb, smaller than in CD_3_OD but large enough that it cannot be confused with an intrinsic shift.

The significant result is that the magnitudes of the equilibrium isotope shifts Δ_eq_ are significantly larger than of the intrinsic isotope shifts Δ_0_ observed in the cationic diacids. The increase in the magnitude of the observed isotope shift, in excess of the intrinsic isotope shift, represents an equilibrium isotope shift Δ_eq_, as listed in Table 3. Therefore, both of these zwitterions exist as a pair of equilibrating tautomers, each with an asymmetric H-bond, exactly as seen with phthalate monoanion. The observed isotope shifts, clearly larger than the intrinsic, are evidence for a tautomeric equilibrium in both these solvents, whether protic or aprotic. Were there no such equilibrium, there would be no additional isotope shift in the zwitterion. Therefore, each of these zwitterions is definitely not present as a single symmetric species. Instead, there are (at least) two species, each with asymmetric H-bonds.

Although an interaction with an asymmetrically located counterion can be responsible for stabilizing an asymmetric H-bond, as in **2** and **3**, this interaction cannot desymmetrize zwitterion **27**, because the quaternary nitrogen lies on a symmetry axis. These experimental results therefore suggest that asymmetry is inherent to all solutions, not through the counterion, but through the disorder of interactions with individual solvent molecules. In principle, the conformational disorder of the alkyl chains might also contribute, but their interactions are much weaker than those of the solvent molecules that are closer to the carboxyls. Those solvent molecules are continuously rearranging their dipole moments, so that the instantaneous stabilization varies with time and with location. This is in contrast to the organized environment found in crystals. The disorder of solvation is a fundamental feature of solutions. It is obvious, but has hardly been explored.

Although these results require a mixture of at least two tautomers, rather than a single symmetric species, we cannot conclude that the H-bond is described by a double-well potential. This may be so, with the instantaneous solvation stabilizing one well more than the other. The alternative is a single-well potential where the instantaneous solvation stabilizes an asymmetric structure, as in Figure 10 (another possibility, a double-well potential where the zero-point energy lies above the barrier, is equivalent to a single-well potential for the purposes of this discussion). As the solvation changes, the hydrogen moves across the H-bond. In each of these structures, the H-bond is asymmetric, and the equilibrium among them can be perturbed by isotopic substitution. These structures can be called **solvatomers** [97], signifying isomers or stereoisomers or (as here) tautomers that differ in solvation (this is a more proper use of the term than an earlier one referring to species that differ in the *type* of solvent molecules [98]. Those do not fit the definition of isomers). Moreover, the solvation is variable, with no unique extent of stabilization but rather an infinity of energy diagrams like Figure 10. The disorder of solvation renders the two donor atoms instantaneously inequivalent, whereupon the hydrogen attaches to the less well solvated donor.

## 12. Generality of Symmetry Breaking by Solvation

All of the above experiments, across a wide range of solvents and across a wide variety of H-bonded species, have found that every one of those species is present as a mixture of asymmetric tautomers. We have failed to detect a symmetric species, although it is possible that those are a low-temperature phenomenon [99,100]. Our experimental results therefore suggest that asymmetry is inherent to all solutions, at least near room temperature or slightly below [101]. The asymmetry is attributed to the inherent disorder of the local environment, which instantaneously solvates one of the basic sites better than the other, stabilizing one tautomer over the other. This is true regardless of whether that solvation is by H-bonding by hydroxylic solvent to carboxylates, by proximity of the counterion to one of the charged donor atoms, or by the orientation of solvent dipoles. This disordered environment due to solvation is quite different from the organized environment found in crystals. 

There are many other cases where the local environment reduces symmetry. One of the most familiar is in the theory of electron or proton transfer, where reorganization energy must be provided to an asymmetric ground-state system in order to achieve a more symmetric configuration that allows the transfer to occur [102]. A classic example is NH_3_ [103], where nitrogen inversion is subject to a double-well potential. In the gas phase, the nitrogen is delocalized between the two wells. If it could be localized in one well, it would rapidly tunnel to the other. However, in an interacting solvent, the nitrogen becomes pyramidal, and the inversion barrier in substituted derivatives can be measured [104]. Other examples are those where the selection rules for IR and Raman intensities break down when the symmetry is reduced by the local solvation environment. Examples include HF_2_^–^ [105,106], H_2_OHOH_2_^+^ [107], CS_2_ [108], I_3_^–^ [109], NO_3_^–^ [110], and aqueous thiourea [111]. A more subtle effect is the effect of solvent on the intensity ratios in the vibronic fine structure of pyrene fluorescence. This correlates with solvent polarity [112], but it is also consistent with the ability of the solvent to disrupt the local symmetry of the molecule and allow otherwise weak transitions. All of these phenomena are worthy of further study to elucidate the role of solvation in breaking symmetry.

## 13. “Strongest” Hydrogen Bond

According to neutron-diffraction studies, some dicarboxylate monoanions are considered to have strong H-bonds, with their H centered between two oxygens [40]. Nevertheless, we have detailed above how we found all such monoanions in solution to be mixtures of tautomers rather than a single, symmetric structure. Yet perhaps none of those monoanions has a sufficiently strong H-bond. Where is the strongest H-bond? One definitive gauge of H-bond strength is Δp*K*_a_, the difference between the first and second acid-dissociation constants [113]. The largest known Δp*K*_a_, 9.54, is for (±)-2,3-di-*t*-butylsuccinic acid (**28**, along with its enantiomer). The large Δp*K*_a_ can be attributed to the steric bulk of the *t*-butyl groups, which favor a conformation (**29** or its enantiomer) that forces the carboxyls into proximity and allows formation of a strong intramolecular H-bond in its monoanion. This can be contrasted with the meso stereoisomer, where the favored conformation (**30**) allows not only the *t*-butyls but also the carboxyls to be apart. 



That Δp*K*_a_ of 9.54 is still puny compared to the 160 kJ-mol^−1^ stabilization energy in FHF^−^ [114]. However, it must be recognized that the latter stabilization is a gas-phase value, so high because bare F^–^ is highly destabilized by confining the negative charge to a small volume. The proper comparison is the Δ*G* or Δ*H* for formation of FHF^–^ in water. Those values are only −2.26 and +6.35 kJ-mol^−1^ (endothermic!), respectively [115]. Therefore, FHF^–^ should not be taken as an example of a strong H-bond even though it is short and symmetric.

The symmetry of the H-bond in the monoanion of **28** was therefore probed using the method of isotopic perturbation. In CD_3_OD, acetone-*d*_6_, and THF-*d*_8_, there is an intrinsic ^18^O-induced isotope shift −Δ_0_ of 22–25 ppb at the carboxyl carbon, and an additional perturbation shift −Δ_eq_ of 8, 14, and 5 ppb, respectively [116]. Therefore, this anion exists in solution as an equilibrating mixture of tautomers, not as a single symmetric species. Moreover, the X-ray crystal structures of five different racemic salts of that monoanion all show asymmetric H-bonds, with the H closer to one oxygen than to the other, even though the O–O distances are only 2.41 Å. Therefore, this short distance cannot be taken as characteristic of a strong H-bond. Instead, the large Δp*K*_a_, of 9.54 is not due to any feature of the H-bond itself but must be attributed to the severe electrostatic repulsion between the two carboxylates in the dianion, which is relieved in the monoanion by inserting a proton between them. Indeed, that repulsion in the dianion is manifested by an O–O distance that is expanded to 2.98 Å [116].

The asymmetry of such H-bonded ions in solution has been attributed to the disorder of the solvation environment and the consequences of their presence as solvatomers, as described above [97]. Why though are they also asymmetric in these crystals? That asymmetry may similarly be attributed to a static asymmetry of the placement of the counterion or to crystal packing forces, which can be recognized in some of those crystals. 

The unusual strength of symmetric H-bonds is often attributed to the resonance stabilization that is maximized with two identical resonance forms. However, the fact that the H-bond in the monoanion of **28** is asymmetric means that the two resonance forms are not identical. Consequently, its H-bond cannot benefit from that maximum of resonance stabilization. Moreover, that asymmetry means that the stabilization associated with two identical resonance forms is not large enough to constrain the H-bond to be symmetric. Even enols of β-dicarbonyl compounds, such as **12**, which are unquestionably resonance-stabilized, do not have symmetric H-bonds. Therefore, resonance does not symmetrize their H-bonds, despite their designation as resonance-assisted [8]. 

If symmetric H-bonds were so stable, we ought to have found some examples. Moreover, if they were so stable, the local solvation environment ought not to have been so capable of disrupting their symmetry. Furthermore, these results deny any relationship between shortness of H-bonds and their strength. Instead, we conclude that H-bonds are simply permissive of short distances. 

## 14. A Serious Critique

Bogle and Singleton proposed an alternative interpretation of the NMR data that we claimed as evidence for asymmetric tautomers [117]. They calculated quasiclassical H trajectories across the highly anharmonic potential-energy surface in isotopically labeled gas-phase hydrogen phthalate monoanion **3** and then averaged the ^13^C NMR shifts over those trajectories. They concluded that an ^18^O label can produce a significant intrinsic isotope shift, whose magnitude is sufficient to account for the results that we had obtained. If so, there is no need to propose equilibrating tautomers. 

We agree that an intrinsic isotope shift can be substantial when there is coupling between a desymmetrizing mode and anharmonic isotope-dependent modes. The remaining question is whether this calculated isotope shift accounts fully for the isotope shifts that we have measured.

A key to analyzing this proposal is that an intrinsic isotope shift ought to be largely independent of temperature. In contrast, the temperature-dependence of isotope shifts was key to Saunders’s evidence for a mixture of carbocations [25]. Likewise, we had found that carboxyl and ipso isotope shifts of aqueous hydrogen phthalate monoanion **3** both decrease with increasing temperature, whereas the carboxyl isotope shift of the corresponding dianion, which must be intrinsic, hardly varies with temperature. Bogle and Singleton seem to accept the disorder of the water environment as strong enough to produce asymmetric ions, but they reject asymmetry in aprotic organic solvents.

To answer this question, it was thus necessary to evaluate the temperature dependence of the isotope shift in an aprotic organic solvent. If the isotope shift is due to perturbation of an equilibrium, it ought to increase at lower temperature. If the isotope shift is due to the desymmetrizing effect of isotopic substitution on a symmetric H-bond, then it may be inferred that it is not necessarily temperature-independent, as asserted above. Instead, we expect that it would decrease at lower temperature, where the vibrational amplitudes decrease and vibrations become more harmonic.

Therefore, we measured the ^18^O-induced ^13^C NMR isotope shifts at the carboxyl and alkene (“ipso”, for parallelism with **3**) positions of mono-^18^O-labeled cyclohexene-1,2-dicarboxylate monoanion **9** [118]. This monoanion was chosen because it had been found to exhibit a large perturbation shift in water [49]. It can be studied in CDCl_3_, as its tetrabutylammonium salt, so as to guarantee solubility. Figure 11 shows the carboxyl region of its ^13^C NMR spectrum.

In Figure 11, the first subscript on label A represents the number of ^18^O atoms attached to a carboxyl carbon, while the second subscript represents the number of ^18^Os attached to the other carboxyl. This assignment is consistent with a shielding due to a heavy atom and with the ^18^O content as measured by mass spectrometry, and it was confirmed by an increase in the A_00_ intensity on adding unlabeled **9**. 

The further splitting arises from a four-bond isotope shift ^4^Δ due to ^18^O in the carboxyl group on the opposite side of the ion. Those splittings can be analyzed as combinations of intrinsic and perturbation shifts, and the perturbation shifts are larger at lower temperature, but the analysis is complicated and the temperature effects are small. 

More revealing are the isotope shifts at the ipso carbons of the mixture of ^18^O isotopologues of **9**. Figure 12 shows their temperature dependence. The five signals, from left to right, are assigned as B_02_, B_01_+B_12_, B_00_+B_11_, B_10_+B_21_, and B_20_, where the first subscript represents the number of ^18^Os on the carboxyl carbon adjacent to the ipso carbon of interest, and the second subscript represents the number of ^18^Os on the opposite carboxyl (the sums represent an unresolvable pair of signals, separated by a negligible intrinsic isotope shift. The relative intensities agree with intensities calculated from the ^18^O distribution measured by mass spectrometry). The key result is that the separations increase with decreasing temperature.

This is strong evidence for a tautomeric equilibrium. The observed separations can be assigned as a perturbation isotope shift ^2^Δ_eq_-^3^Δ_eq_. The temperature dependence of the isotope shifts suggests that the dominant origin of those isotope shifts is the perturbation of an equilibrium by ^18^O substitution. That equilibrium must be among tautomers that differ in whether the proton resides on the ^18^O-labeled carboxyl or on the unlabeled one, just as was inferred for **3**. Although the H-bond may be intrinsically symmetric, with a single-well potential, we conclude that the asymmetry arises from the disorder of solvation and the presence of solvatomers. Therefore, the H-bond in the monoanion of **9** is asymmetric not only in water but also in chloroform. 

We therefore dispute the conclusion of Bogle and Singleton that the isotope shift is intrinsic [117], and we reaffirm the conclusion that dicarboxylate monoanions such as **9** are asymmetric not only in aqueous media but also in organic solvents [118]. We do not deny that coupling between a desymmetrizing mode and anharmonic isotope-dependent modes can contribute to the isotope shift. The question is whether this is the dominant contribution or whether the dominant contribution is the perturbation of an equilibrium among tautomers by the instantaneous local environment. The temperature dependence that we observe suggests the latter. 

We do recognize that a remaining question is whether the observed temperature dependence can be reproduced by calculations of the trajectory of hydrogen motion across the potential-energy surface of a H-bonded monoanion. To the extent that lower temperature decreases the amplitudes of the motions and the mixing with anharmonic modes, we expect that a calculated isotope shift would decrease at lower temperature. If so, this would be inconsistent with our observation of an increased isotope shift at lower temperature. We had recommended a calculation of the temperature dependence of the intrinsic isotope shift due to the coupling of anharmonic vibrations, but as yet none has been forthcoming. Therefore, we conclude that the observed isotope shifts mean that the H-bond is asymmetric, and we doubt that they arise from the desymmetrizing effect of a coupling of anharmonic vibrations. 

## 15. ODO H-Bonds vs. OHO

As a further test of that alternative interpretation of our NMR data, we studied how those data would be affected by deuteration in the H-bond. The aim was to compare the isotope shifts in two diacid monoanions with an ODO H-bond to those same monoanions with an OHO H-bond. If the isotope shifts are due to the desymmetrizing effect of isotopic substitution on a symmetric but anharmonic potential-energy surface, that effect ought to be smaller with heavier deuterium, whose motion is less anharmonic. Alternatively, if the isotope shifts are due to the perturbation of an equilibrium among asymmetric H-bond tautomers, that equilibrium might become more unbalanced with OD, because the isotope shift is expected to increase with D [45], owing to a larger ^18^O IE on the acidity of an OD acid [44,119]. 

We therefore measured ^18^O-induced isotope effects on the ^13^C NMR chemical shifts of the tetrabutylammonium salt of cyclohexene-1,2-dicarboxylate monoanion **9** in chloroform-*d* and on the ^19^F NMR chemical shifts of difluoromaleate monoanion **31** in D_2_O [120]. Figure 12 above shows all the ^13^C NMR signals of the ipso carbons in **9**-*h*-^18^O. Figure 13 shows the temperature dependence of the isotope shifts at the ipso carbons of both **9**-*h*-^18^O and **9**-*d*-^18^O, evaluated as half the separation between B_01_+B_12_ and B_10_+B_21_ peaks. Just as for **9**-*h*-^18^O, the isotope shifts of **9**-*d*-^18^O are larger at lower temperature. More diagnostic is the observation that the isotope shifts are larger with deuterium in the H-bond. It is thereby concluded that these behaviors are consistent with the perturbation of an equilibrium among asymmetric tautomers and inconsistent with isotope-induced desymmetrization on a symmetric potential-energy surface.

The ^19^F NMR spectra of the tetrabutylammonium salts of 15:80:5 mixtures of un-, mono-, and di-^18^O-labeled difluoromaleate monoanions (**31**-*h*-^18^O*_n_* and **31**-*d*-^18^O*_n_*, *n* = 0, 1, 2) in D_2_O at 20 °C are shown in Figure 14. In each spectrum, the two central signals, one stronger the other weaker, can be assigned to the ^18^O_0_ and ^18^O_2_ isotopologues, respectively. Notice also that the signals of **31**-*h*-^18^O*_n_* in Figure 14a are approximately twice as broad as those of **31**-*d*-^18^O*_n_* in Figure 14b. This can be attributed to unresolved ^4^*J*_HF_, larger than ^4^*J*_DF_. 

It is noteworthy that even in D_2_O, **31**-*h* retains its H. This H does not exchange out of its H-bond because monoanion **31** is so weak an acid that it does not lose its H without added strong base. Nor is **31** sufficiently basic to be converted to its diacid, from which that H could have been removed. 

All the chemical-shift separations for **31**-*h*-^18^O*_n_* and **31**-*d*-^18^O*_n_* in Figure 14 increase with decreasing temperature. Again, the increases can be attributed to the perturbation of an equilibrium between tautomers that differ in whether the H or D is on ^16^O or ^18^O. 

Most revealing is the recognition that the outer four ^19^F NMR signals in each part of Figure 14, which can be assigned to **31**-^18^O_1_, can be recognized as an AB spin system [121]. Therefore, the two Fs are magnetically inequivalent and can split each other. In contrast, the disodium salt of that same 15:80:5 mixture shows only a singlet. Therefore, the inequivalence is not simply due to the presence of ^18^O but must be due to the H or D in the H-bond. Because the two Fs are inequivalent, the H or D cannot be located symmetrically with respect to those Fs. Instead, the H-bond must be asymmetric!

## 16. Steric Compression

Bulky substituents force the two oxygens of an OHO H-bond closer together and favor a single-well potential [121], as do electron-withdrawing substituents at the central carbon [122]. Therefore, we have undertaken to investigate the symmetry of the H-bonds in two sterically encumbered enols, 4-cyano-2,2,6,6-tetramethyl-3,5-heptanedione enol (**32**) and nitromalonamide enol (**33**).



The O–O distance in crystalline **32** is an unusually short 2.393 Å and with C−O distances of 1.273 and 1.274 Å and C–C distances of 1.430 and 1.432 Å [123]. Moreover, X-ray crystallography shows an almost perfectly *C*_2v_-symmetric structure. However, according to neutron-scattering measurements, the H although nearly centered between the two oxygens resides in a low-barrier double-minimum potential. For **33**, the O–O distance is 2.391 Å, with the H located asymmetrically between the two oxygen atoms but in a single-well potential, with a calculated barrier to H transfer of only 0.15 kcal/mol [124]. A low barrier, below the zero-point energy, is consistent with a mixture of tautomers, as evidenced by only a small ^2^Δ(C) on replacing H in the H-bond by D [125].

Again, we used the NMR method of isotopic perturbation to address this question, again with statistical mixtures of ^18^O*_n_* isotopologues (*n* = 0, 1, 2) [126]. Figure 15a shows the carbonyl region of the ^13^C NMR spectrum of **32**. There are four signals, the outermost of which can be assigned to the two carbons of **32**-^18^O_1_ and the innermost to **32**-^18^O_0_ and **32**-^18^O_2_, separated by an intrinsic shift of 26 ppb. The smaller separation, between **32**-^18^O_1_ and **32**-^18^O_0_ or between **32**-^18^O_2_ and **32**-^18^O_1_, is −12.1 ppb at 21.6 °C, but −14.6 ppb at −20.0 °C. Therefore, this is a perturbation shift that is evidence for a mixture of tautomers and an asymmetric H-bond, as for other enols.

Figure 15b shows the carbonyl region of the ^13^C NMR spectrum of **33**. There are only two signals, of the same linewidth as for **32**. According to the ^18^O content as measured by mass spectrometry, the more intense signal is assigned to the carbon attached to ^16^O and the less intense to the carbon attached to ^18^O. The separation between the two signals is 18.6 ppb at both 21.6 °C and −20 °C. Because there are only two signals, not the four for **32**, there is no detectable isotope shift due to perturbation of an equilibrium. Moreover, the temperature independence is evidence against a perturbation shift. Instead, the separation must be assigned as an intrinsic isotope shift. Therefore, we conclude that the H-bond in enol **33** is symmetric, with a centered hydrogen.

The surprising result is that **33** is a single symmetric species, not a mixture of tautomers [126]. Indeed, we claim that **33** is the first definite example besides FHF^–^ and H_2_OHOH_2_^+^ of a symmetric H-bond near room temperature. One distinguishing feature of **33** is that it is a neutral species, whereas the previous examples of asymmetric dicarboxylate monoanions and ammonium cations are ions. Solvation is more important for those ionic species, whereas solvation of the nonionic **33** is weaker and less susceptible to solvent disorder. This example is worthy of further investigation, perhaps by femtosecond two-dimensional infrared spectroscopy [127].

## 17. Conclusions

Isotopic perturbation is a powerful NMR method for determining the symmetry of H-bonds. We have used this method to investigate a wide variety of molecules in solution, including various dicarboxylate monoanions, aldehyde enols, diamines, enamines, acid–base complexes, and two sterically encumbered enols. Among all of these, we have found only one example of a symmetric H-bond, in nitromalonamide enol (**33**). We have attributed the near universality of asymmetric H-bonds to the presence of solvatomers, isomers that differ in their solvation.

We deplore the widespread invocation of short, strong, symmetric, low-barrier H-bonds as reflective of a special stability or of enhanced catalytic activity. We raise the question of whether there is any relationship among symmetry, shortness, and strength of H-bonds. Yes, there is undoubtedly a relationship between symmetry and shortness, because decreasing the distance between donor atoms must eventually decrease the barrier to hydrogen motion, until a double-well potential becomes a single-well one. However, observations of short H-bonds are usually not accompanied by evidence for their symmetry. Indeed, the rarity of symmetric H-bonds in solution suggests that they have no special stability. 

As for a relationship between shortness and strength, such a relationship has become a common supposition, leading to numerous claims for strong H-bonds when the only evidence is a short heavy-atom distance. One reason that short H-bonds are believed to be unusually strong seems to be an influential graph relating H-bond strength to heavy-atom distance [128]. The graph does not show a linear correlation. Instead, it shows a distinct jump, almost a discontinuity, between short, strong bonds in the gas phase and long, weak ones in solution. The apparent relation arises simply because all the weak H-bonds are neutral species, with no special stabilization, whereas all the strong ones (with one dubious exception) are gas-phase ions, which are stabilized by an enhanced electrostatic attraction. Therefore, we deny any relation between shortness and strength. 

Is there any relation between symmetry and strength, as implied by the mantra of strong, low-barrier H-bonds? Because nearly all the H-bonds that we have investigated, with only one exception, are asymmetric in solution, we conclude that there is no H-bond that is stabilized by symmetry per se nor any special stabilization associated with symmetric, short, or low-barrier H-bonds. If symmetric or low-barrier H-bonds were so stable, they ought to be more common and we ought to have found more of them. Besides, if they were so stable, the local solvation environment should not be capable of disrupting their symmetry. 

Unfortunately, many researchers ascribe a special significance to short, strong, symmetric, or low-barrier H-bonds, often without any acknowledgment of contrary evidence. Table 4 lists a few recent publications that seem to support such exaggeration. They do not deserve explicit citation.

In conclusion, we deny that there is any relation between shortness of H-bonds and H-bond strength or between symmetry and strength. We urge that scientists stop invoking a special importance for short, strong, or low-barrier H-bonds.

## Figures and Tables

**Figure 1 molecules-28-04462-f001:**
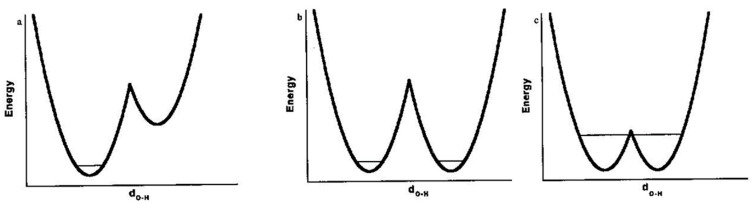
Potential-energy curves for AHB H-bonds: (**a**) asymmetric double-well potential, (**b**) symmetric double-well potential, (**c**) low-barrier H-bond.

**Figure 2 molecules-28-04462-f002:**
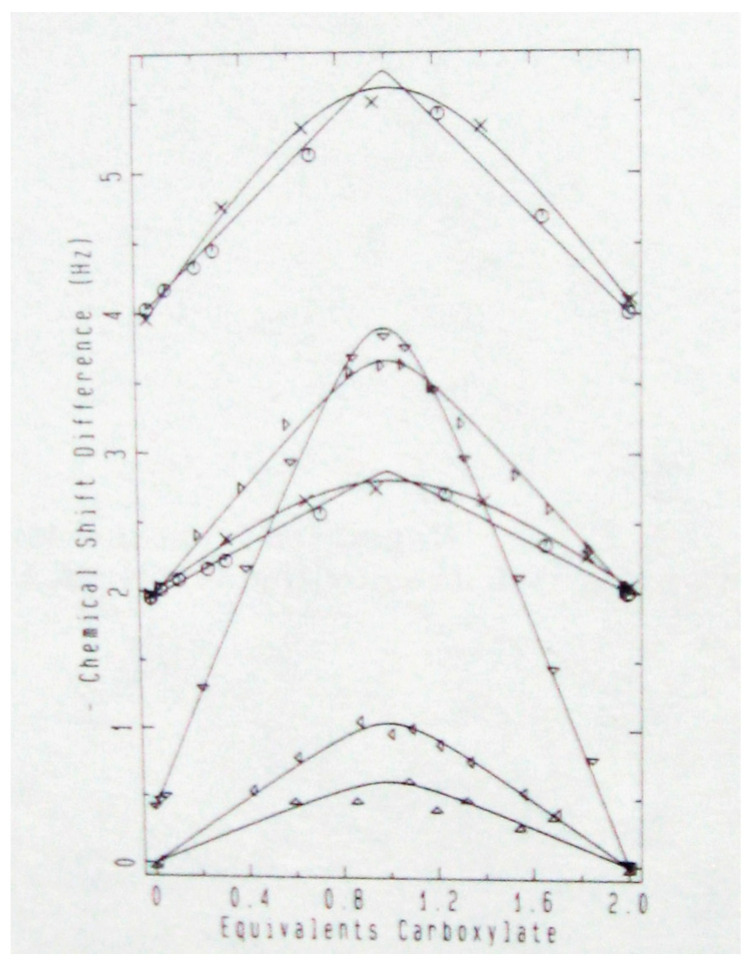
Observed isotope shift vs. extent of neutralization: succinic acid (x), maleic acid (o), phthalic acid: carbonyl (Δ), ipso (∇), meta (Δ), and ortho (Δ). Reprinted with permission from *JACS* ***111*,** p. 8011. Copyright 1989 American Chemical Society.

**Figure 3 molecules-28-04462-f003:**
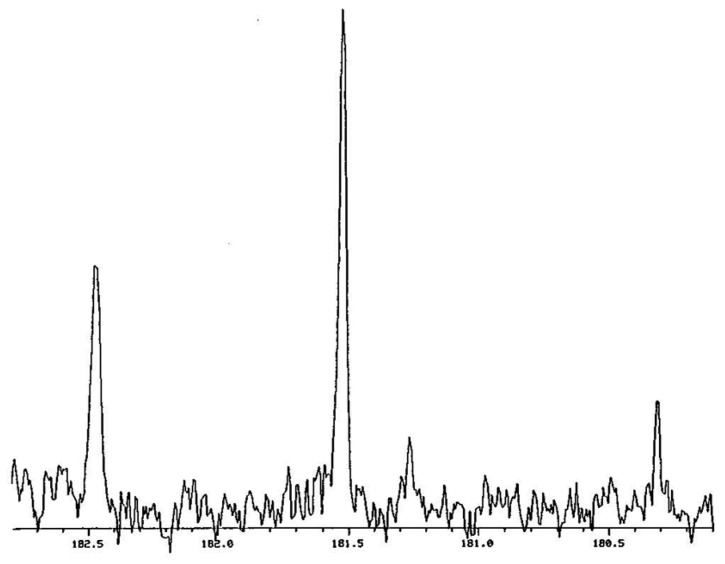
Carbonyl region of ^1^H- and ^2^H-decoupled ^13^C NMR spectrum of a mixture of **12**, **12**-*d*_1_, and **12**-*d*_2_ in CDCl_3_. Reprinted with permission from *JACS* ***120*,** p. 12644. Copyright 1998 American Chemical Society.

**Figure 4 molecules-28-04462-f004:**
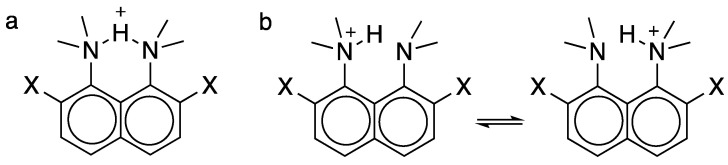
(**a**) Symmetric and (**b**) asymmetric H-bonds of protonated 1,8-bis(dimethylamine)naphthalenes.

**Figure 5 molecules-28-04462-f005:**
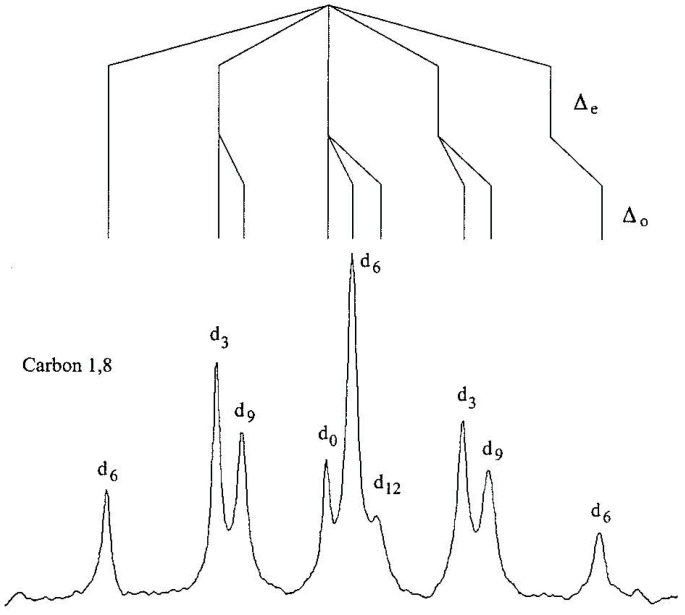
C1,8 region of the ^13^C NMR spectrum of **21**·HSCN in DMSO-*d*_6_. Reprinted with permission from *JACS 123***,** p. 6524. Copyright 2001 American Chemical Society.

**Figure 6 molecules-28-04462-f006:**
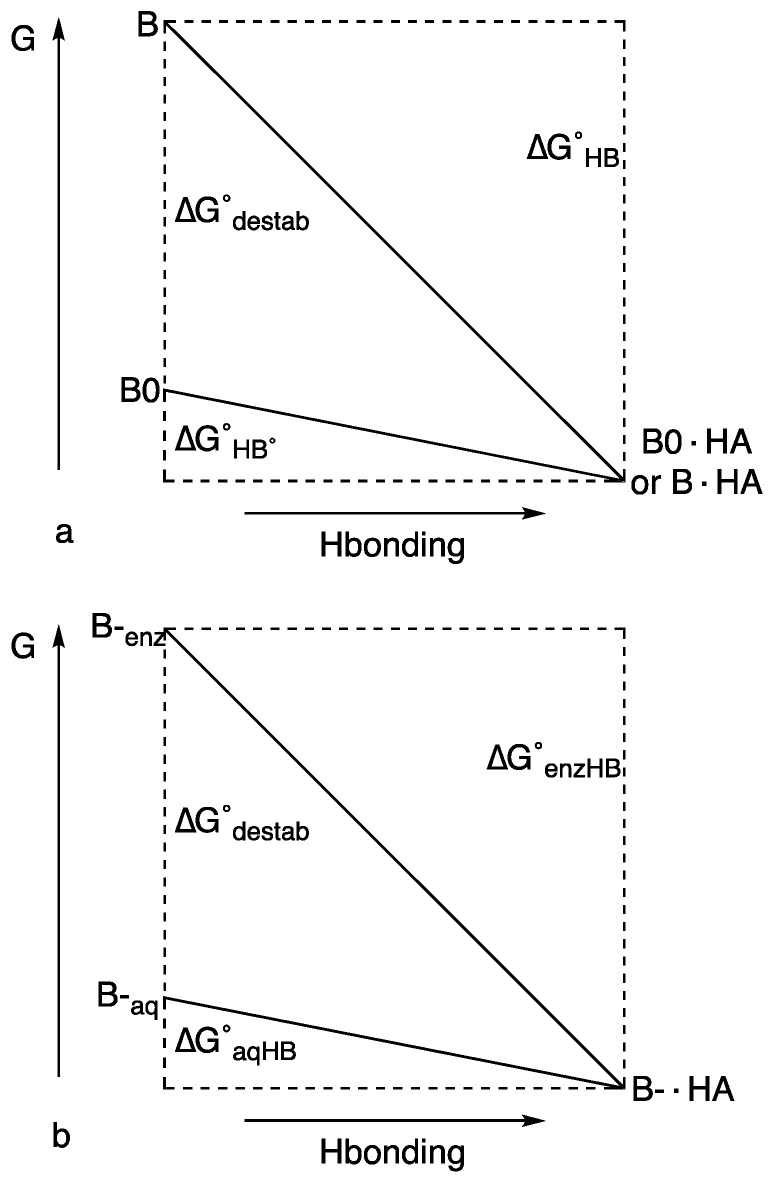
H-bond energy and strain relief (**a**) in protonated bis(dimethylamino)naphthalenes **21** and **22** and (**b**) in enzymes. Reprinted with permission from *JACS* ***123***, p. 6526. Copyright 2001 American Chemical Society.

**Figure 7 molecules-28-04462-f007:**
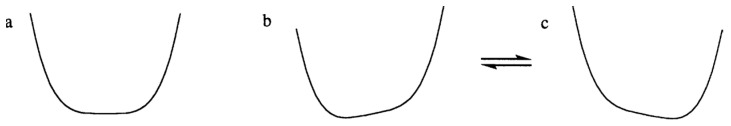
Possible effects of the local solvation environment on a single-well potential: (**a**) without solvation, (**b**,**c**) as solvent undergoes rapid reorganization. Reprinted from “Symmetry of NHN hydrogen bonds in solution”, *J. Mol. Struct*, Vol. 644, pp. 1–12, by Charles L. Perrin & Brian K. Ohta, p. 9, Copyright 2003, with permission from Elsevier.

**Figure 8 molecules-28-04462-f008:**
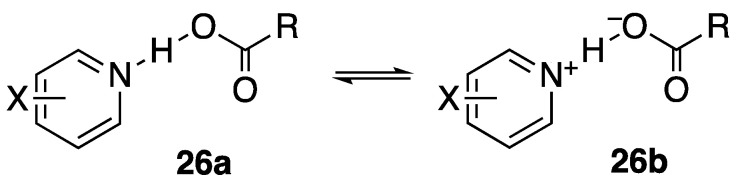
Scan for a basicity match between a pyridine and dichloroacetate.

**Figure 9 molecules-28-04462-f009:**
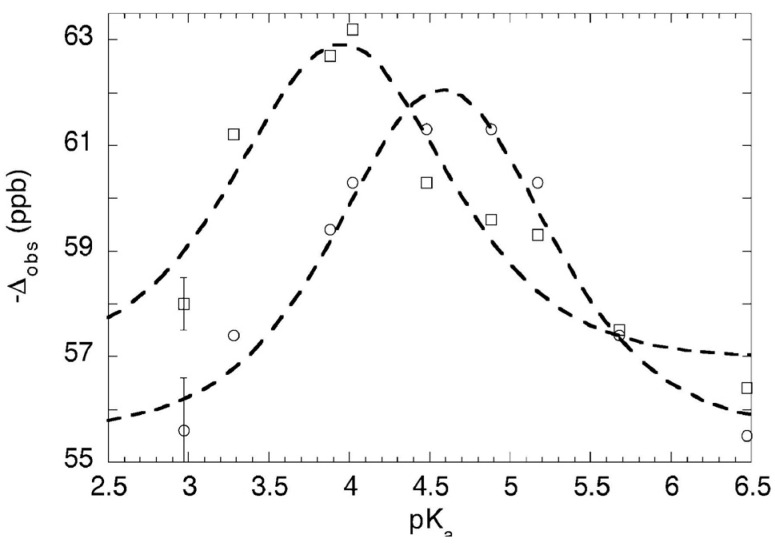
^13^C isotope shift Δ vs. pyridine p*K*_a_ in pyridine-dichloroacetic acid complexes: (o) at –43 °C, (⊡) at –81 °C, with smooth curves fitted to Equation (7). Reproduced from Ref. [96] (*Chem. Commun*., **2010**, p. 482) with permission from the Royal Society of Chemistry.

**Figure 10 molecules-28-04462-f010:**
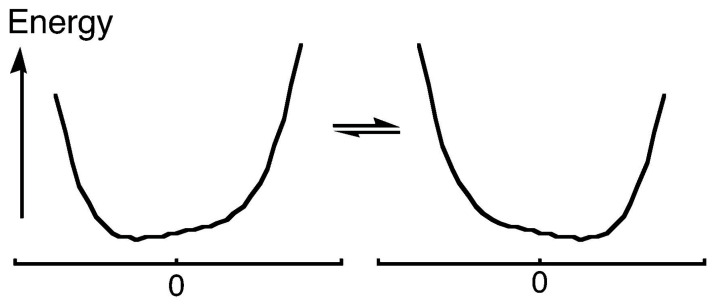
Equilibrating H-bonded AHB solvatomers, each with a single-well potential describing energy vs. bond-distance difference *d*_AH_−*d*_HB_. Reprinted with permission from *JACS* ***128***, p. 11823. Copyright 2006 American Chemical Society.

**Figure 11 molecules-28-04462-f011:**
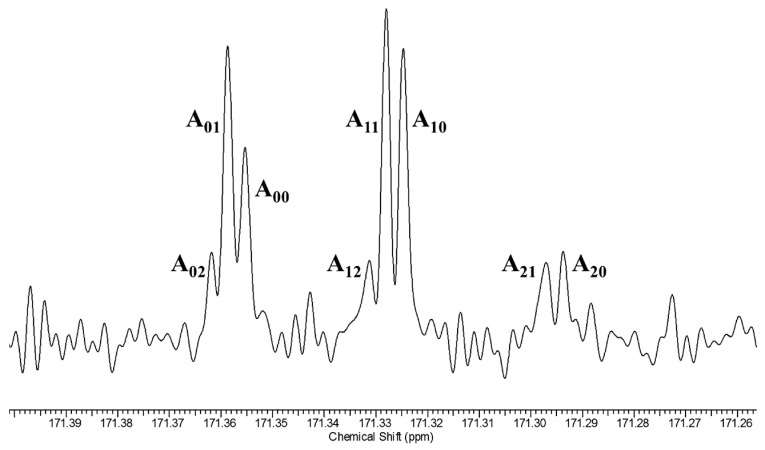
^13^C NMR spectrum of the carboxyl region of a mixture of ^18^O-labeled isotopologues of **9** in CDCl_3_ at 293.9 K. Reprinted with permission from *JACS 126*, p. 4358. Copyright 2014 American Chemical Society.

**Figure 12 molecules-28-04462-f012:**
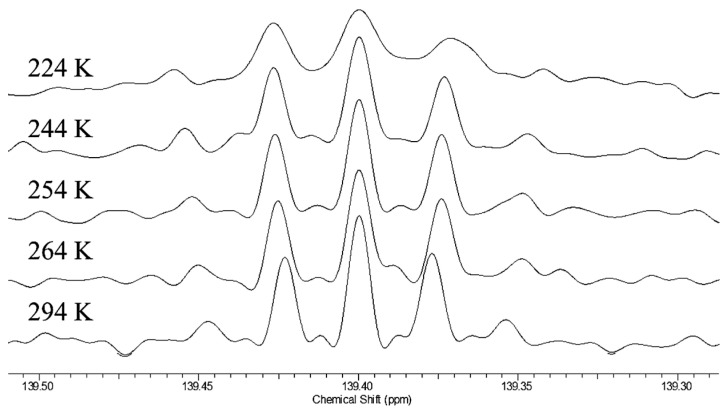
Temperature dependence of the ^13^C NMR signals of the ipso carbons in ^18^O isotopologues of **9**. Reprinted with permission from *JACS 126*, p. 4359. Copyright 2014 American Chemical Society.

**Figure 13 molecules-28-04462-f013:**
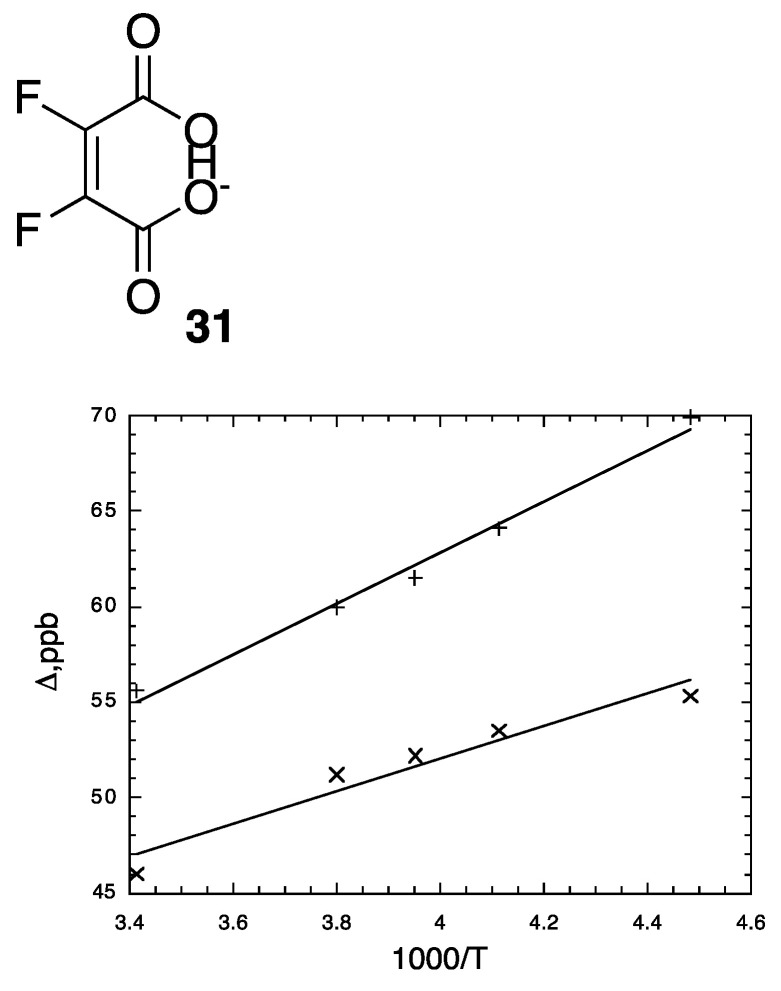
Temperature-dependence of isotope shifts of ipso carbons in **9**-*h*-^18^O (x) and **9**-*d*-^18^O (+) in CDCl_3_. Reprinted with permission from *JACS 141***,** p. 17281. Copyright 2019 American Chemical Society.

**Figure 14 molecules-28-04462-f014:**
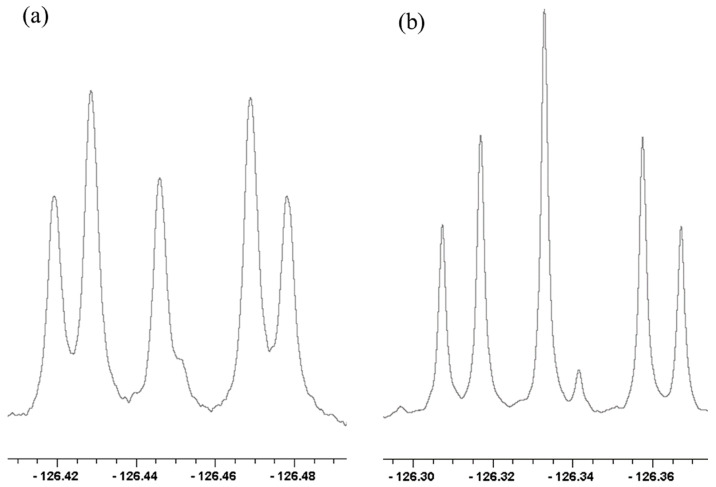
^19^F NMR spectra of (**a**) **31**-*h*-^18^O*_n_* and (**b**) **31**-*d*-^18^O*_n_*, *n* = 0,1,2 in D_2_O at 20 °C. Reprinted with permission from *JACS 141*, p. 17282. Copyright 2019 American Chemical Society.

**Figure 15 molecules-28-04462-f015:**
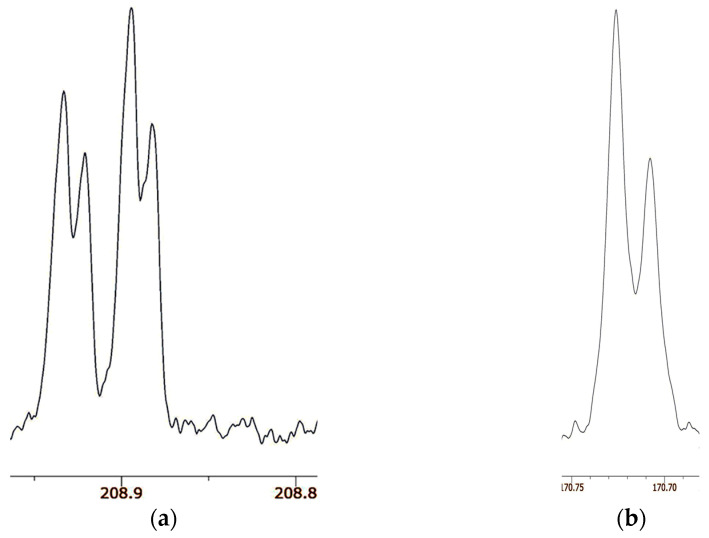
Carbonyl region of ^13^C NMR spectra of (**a**) **32** in CDCl_3_ and (**b**) **33** in DMSO-*d*_6_. Reprinted with permission from *JACS 141***,** p. 4105. Copyright 2019 American Chemical Society.

**Table 1 molecules-28-04462-t001:** ^18^O-induced isotope shifts (ppb) of dicarboxylic acid monoanions in DMSO-*d*_6_.

Anion	Signal	Δ_o_−Δ_HA−_
**3**	C=O	1
**3**	ipso	11
**4**	C=O	2
**4**	C1,2	16
**5**	C=O	6
**5**	ipso	>9
**6**	C=O	0
**6**	C2,3	42
**7**	C=O	0
**8**	C=O	3
**9**	C=O	3
**9**	C1,2	27

**Table 2 molecules-28-04462-t002:** ^13^C NMR chemical shifts and significant isotope shifts of ^(a)^ **21**·HSCN and ^(b)^ **22**·HBF_4_ in DMSO-*d*_6_ [74].

Carbon	Δ_o_, ppb	Δ_eq_, ppb
1,8 ^a^	−29	−120
2,7 ^a^	<5	35
3,6 ^a^	<5	18
4,5 ^a^	−13	32
9 ^a^	18	0
NCH_3_ ^a^	−78	<5
2,7 ^b^	<5	47
4,5 ^b^	−12	14
9 ^b^	15	0
NCH_3_ ^b^	−80, −25	<5

**Table 3 molecules-28-04462-t003:** ^18^O-induced ^13^C isotope shifts (ppb) of zwitterions **27** [97].

R	Solvent	Carbon	−Δ_obs_	−Δ_0_	−Δ_eq_
*n*-butyl	CD_3_OD	carboxyl	40	25	15
*n*-butyl	CD_3_OD	ipso	35	4	31
*n*-octyl	CD_3_OD	carboxyl	40	25	15
*n*-octyl	CD_3_OD	ipso	39	2	37
*n*-octyl	CD_2_Cl_2_	carboxyl	28	26	2
*n*-octyl	CD_2_Cl_2_	ipso	12	<5	>7

**Table 4 molecules-28-04462-t004:** Some recent publications that ascribe a special significance to short, strong, symmetric, or low-barrier H-bonds.

Exploring short strong hydrogen bonds engineered in organic acid molecular crystals for temperature dependent proton migration behaviour using single crystal synchrotron X-ray diffraction (SCSXRD)
Studying the hydrogen atom position in the strong-short intermolecular hydrogen bond of pure and 5-substituted 9-hydroxyphenalenones by invariom refinement and ONIOM cluster computations
Core level spectroscopies locate hydrogen in the proton transfer pathway—identifying quasisymmetrical hydrogen bonds in the solid state
An N/H/N low-barrier hydrogen bond preorganizes the catalytic site of aspartate aminotransferase to facilitate the second half reaction
True-atomic-resolution insights into the structure and functional role of linear chains and low-barrier hydrogen bonds in proteins

## Data Availability

Not applicable.

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
