# Peer review of "Symmetry of Hydrogen Bonds: Application of NMR Method of Isotopic Perturbation and Relevance of Solvatomers"

_molecules, 2023, doi:10.3390/molecules28114462_

Round 1

Reviewer 1 Report

This manuscript "Symmetry of Hydrogen Bonds: Application of NMR Method of Isotopic Perturbation and Relevance of Solvatomers " provides a comprehensive overview of the use of isotopic perturbation as a powerful NMR method for determining the symmetry of H-bonds. The authors have used this method to investigate various dicarboxylate monoanions, aldehyde enols, diamines, enamines, acid-base complexes, and two sterically encumbered enols, and have found only one example of a symmetric H-bond. The authors attribute the near universality of asymmetric H-bonds to the presence of solvatomers, isomers that differ in their solvation. The authors then go on to discuss the common supposition that there is a relationship between symmetry, shortness, and strength of H-bonds, and provide evidence to deny this relationship. They also provide a table of recent publications that ascribe a special significance to short, strong, symmetric, or low-barrier H-bonds.
Overall, this is a well-written and comprehensive manuscript that provides an interesting and informative overview of the use of isotopic perturbation to determine the symmetry of H-bonds. The authors have provided a thorough discussion of the relationship between symmetry, shortness, and strength of H-bonds, and have provided evidence to deny this relationship. The manuscript is well-structured and easy to follow, and the authors have provided a clear conclusion. The only minor revision that could be made is to provide more detail on resently results.
I suggest that further research be described to investigate the relationship between с, shortness, and H-bond strength [10.3390/sym12111924]. This could include exploring the effects of solvation [10.1038/s41598-023-30089-x] on the symmetry of H-bonds and the effects of substituents on the strength of H-bonds. Additionally, further should be described to determine the role of symmetry in pressure-induced hydrogen bond reconstruction [10.1016/j.cplett.2023.140572, 10.1016/j.molliq.2022.120525].
Finally, a note about formatting: all figures should be updated to ensure legibility and consistency in the style of the axes and labels. For example, the labels on Graphs 2, 3, and 9 are illegible. Utilize a tool to digitize (for example WebPlotDigitizer) the graphs and obtain the data, then reproduce them following these suggestions.

P.S. Could you please change the Roman numerals for literature to Arabic numerals?

Scientific sound corrections (not essential, but will help make your manuscript clear to understand):

"so that these interactions can become quite strong" -->  "making these interactions quite strong"

"Fortunately there is an NMR method that is applicable" --> "Fortunately, an NMR method is applicable"

"There are several other articles touting" --> "Several other articles are touting"

"The results on hydrogen maleate and phthalate were" -->  "The hydrogen maleate and phthalate results were"

"but at any instant the H-bond itself is asymmetric" --> "but
the H-bond itself is asymmetric at any instant"

"Bi-dentate metal -diketonates are firmly believed to be symmetric, according to crystal structures"-->"according to crystal structures  Bi-dentate metal diketonates are firmly believed to be symmetric"

"even though N and O are not the same" --> " even though N and O are different"

" The two oxygens must have identical basicity, and they are forced close to each other, so that a symmetric H-bond is possible. " -->  "The two oxygens must have identical basicity and are forced close to each other, making a symmetric H-bond possible."

"If symmetric H-bonds were so stable, we ought to have found some examples." --> "We ought to have found some examples if symmetric H-bonds were so stable."

"tautomers by the instantaneous local environment. " -->  "tautomers by the immediate local environment."

"This is strong confirmatory evidence" -->  "This is confirmatory solid evidence"

" 1.2 Å , considerably greater than the usual 1.0 Å" -->  ".2 Å , considerably more significant than the usual 1.0 Å "

"We ought to have found some examples" --> "We should have found some examples"

Author Response

Scientific sound corrections (not essential, but will help make your manuscript clear to understand):

"so that these interactions can become quite strong" -->  "making these interactions quite strong"

changed

"Fortunately there is an NMR method that is applicable" --> "Fortunately, an NMR method is applicable"  

changed

"There are several other articles touting" --> "Several other articles are touting"  

 changed

"The results on hydrogen maleate and phthalate were" -->  "The hydrogen maleate and phthalate results were"            disagree

"but at any instant the H-bond itself is asymmetric" --> "but the H-bond itself is asymmetric at any instant"  disagree

"Bi-dentate metal b-diketonates are firmly believed to be symmetric, according to crystal structures"-->"according to crystal structures  Bi-dentate metal diketonates are firmly believed to be symmetric"        

changed

"even though N and O are not the same" --> " even though N and O are different"           

changed

" The two oxygens must have identical basicity, and they are forced close to each other, so that a symmetric H-bond is possible. " -->  "The two oxygens must have identical basicity and are forced close to each other, making a symmetric H-bond possible."         disagree

"If symmetric H-bonds were so stable, we ought to have found some examples." --> "We ought to have found some examples if symmetric H-bonds were so stable." disagree

"tautomers by the instantaneous local environment. " -->  "tautomers by the immediate local environment."            disagree

"This is strong confirmatory evidence" -->  "This is confirmatory solid evidence"    disagree

" 1.2 Å , considerably greater than the usual 1.0 Å" -->  ".2 Å , considerably more significant than the usual 1.0 Å "            disagree

"We ought to have found some examples" --> "We should have found some examples"    disagree

Reviewer 2 Report

This is an excellent review solving hot debated questions about tautomerism vs. symmetric hydrogen bonds.  However, a couple of papers not by the author could possibly be useful in this discussion:

J. Phys. Org. Chem., 4 (1991) 225-232. and Magn.Reson.Chem.  46 (2008) 726-9.

Author Response

This is an excellent review solving hot debated questions about tautomerism vs. symmetric hydrogen bonds.  However, a couple of papers not by the author could possibly be useful in this discussion:

J. Phys. Org. Chem., 4 (1991) 225-232. and Magn.Reson.Chem.  46 (2008) 726-9

both now cited

Reviewer 3 Report

The review is well written. The spectra present and highlight what the author wants to highlight. The accuracy and commitment of the author is highlighted by the careful search of the references that turn out to be more than 150.Although I would have made the conclusions a little longer.

Author Response

The review is well written. The spectra present and highlight what the author wants to highlight. The accuracy and commitment of the author is highlighted by the careful search of the references that turn out to be more than 150. Although I would have made the conclusions a little longer.  

expanded slightly